# Deciphering Preferences for Shelter Volume and Distribution by Coral Reef Fish, Using Systematic and Functional Grouping

Tamar Shabi [1,2], Yaron Ziv [3], Reuven Yosef [1] and Nadav Shashar [1,3,*]

1 Marine Biology and Biotechnology Program, Department of Life Sciences, Eilat Campus, Ben Gurion University of the Negev, Beer Sheva 84105, Israel; tamar.shabi3@gmail.com (T.S.); ryosef60@gmail.com (R.Y.)
2 The Interuniversity Institute for Marine Science, Eilat 88000, Israel
3 Faculty of Natural Sciences, Department of Life Sciences, Ben Gurion University of the Negev, Beer Sheva 84105, Israel; yziv@bgu.ac.il
* Correspondence: nadavsh@bgu.ac.il

**Abstract:** Global degradation of coral reefs is reflected in the destruction of shelters in various environments and threatens the stability of marine ecosystems. Artificial shelters offer an alternative, but their design could be more challenging due to limited knowledge regarding desired inhabitants' shelter characteristics and preferences. Investigating these preferences is resource-intensive, particularly regarding small shelters that mimic natural reef conditions. Furthermore, for statistical analysis in small shelters, fish abundance may need to be higher. We propose a method to characterize the species-specific shelter preferences using low-volume data. During a study conducted from January 2021 to April 2022, round clay artificial shelters (RAS) were deployed on an abandoned oil pier to examine a coral reef fish community. We recorded 92 species from 30 families and grouped them into systematic (families) and functional (dietary group) classes. Grouping enabled us to examine each group's preference, while crossing these group preferences revealed species-specific preferences, which matched field observations. This approach proved effective in profiling the shelter preferences of 17 species while having limited resources. These profiles may later allow the establishment of ecological-oriented artificial reefs. Moreover, this method can be applied to other applications using other shelter designs, sizes, and research sites.

**Keywords:** artificial reefs; restoration; shelter characteristic; Red Sea; shelter design

## 1. Introduction

Coral reefs are considered an essential part of marine ecosystems due to their biodiversity and their geomorphological contributions, such as preventing erosion and protecting coastlines from hurricanes and tropical storms [1,2]. Coral reefs also play a crucial role in maintaining the stability of the marine environment [3–5]. However, the global distribution area and quantity of coral reefs are decreasing [6–8]. Habitat degradation can be defined as a "change in states between one where the provision of resources leads to an ecosystem with high complexity and species diversity, to a state where the resources do not support communities of high diversity". The lack of suitable shelters for reef fish can threaten the stability of the entire marine ecosystem [5,8,9] and lead to the local extinction of highly specialized species [7,10]. Many researchers believe that human intervention is necessary to maintain the stability of the marine environment, where coral reefs suffer substantial damage [11].

The active restoration approach [12–14] suggests the creation of artificial shelters for reef fish [15]. With a proper design, these artificial shelters can mimic natural reefs, enabling fish to engage in key processes vital for their survival [11]. Moreover, similar to natural reefs, the appropriate substrate can lead to the development of natural fauna on and within the shelters. A recent study [16] assessed the success of artificial shelters, demonstrating that artificial reefs (AR) can offer the same functions and benefits to the marine environment

as natural reefs. ARs were found to enhance fish abundance, improve habitat abundance or coral cover, preserve target species, mitigate stressors, provide a coral nursery habitat for source populations, and address socio-cultural and economic values [16]. Higgins et al. (2022) [16] also observed that the use of ARs in diverse habitats can yield benefits for both benthic and pelagic communities by reducing anthropogenic pressure on natural habitats [17] and offering protection from predators and human disturbance.

Furthermore, ARs can be used for mariculture [11]. Various methods have been implemented, including the construction of artificial coral reefs with different configurations [17,18], the submersion of various artificial structures [19], or the introduction of artificial shelters mimicking reef structures [15]. The aim is to create an alternative habitat for organisms reliant on coral reefs' unique and complex structure, such as fish and motile invertebrates.

Active restoration is gaining popularity, and significant research has been conducted. However, researchers and practitioners still face a major challenge in attempting to create optimal designs for ARs, as predicting fish distribution and shelter choice proves difficult. It was suggested that individuals are likely to choose habitats where their chances of success (survival, fitness [20]) are greatest [21,22]. In coral reefs, researchers are also striving to understand which habitats are considered ideal for reef fish, i.e., which facilitate key processes, guilds, and niches. Some studies examine the preference of the reef fish population in a specific area or conclude specific groups, such as systematic or functional [4,23–26].

Several studies have focused on general characteristics that are beneficial for many reef fish species, such as relative size or the ratio between fish size and shelter size [5,27–30], complexity [5,31,32], number of holes [24,27,29,33], connectivity between shelters [34,35], etc. Specific features of ARs determine the presence of fish communities [26]. However, concluding a specific location and shelter design may not apply to other geographical sites or designs, as different species are likely to choose different shelters based on their own physical and behavioral needs. Moreover, observed fish communities encompass many species with different behavioral patterns, predation, and habitat characteristics needs [36,37]. These features ultimately have different effects on the disturbance of individuals of each species. Consequently, inferences about fish preferences should be made for each species separately rather than considering the entire community in the area [36,37]. Focusing on the preferences of each predominant species in the area can aid in designing a more appropriate AR, leading to greater success in restoring damaged coral reefs. However, characterizing the preferences of individual species in the study area can be challenging and requires a significant investment of resources. Additionally, small shelters, common in natural reefs [24,38], often exhibit inherently low fish numbers, posing challenges for statistical tests due to the limited data.

Using specially designed small round artificial shelters (RAS), we examined fishes' preferences to specific shelters. We focused on examining two commonly studied characteristics of reef fish shelters: size [5,27–30,39] and spatial distribution [34,35]. Moreover, we looked for commonalities between fishes of different species but of similar behavioral-feeding modes. This was implemented by placing the shelter on special balcony-like structures on an abandoned oil jetty. The use of abandoned oil rigs for artificial reef deployment is a well-known practice worldwide ('Rigs-to-reefs'), wherein instead of decommissioning unused oil rigs, they are repurposed into artificial reefs. Jetties, commonly found in submerged structures globally, offer similar potential benefits to the marine environment [40,41].

To uncover reef fish preferences for specific shelters, we focused on examining two commonly studied characteristics of reef fish shelters: size [5,27–30] and spatial distribution [34,35]. In these, we hypothesized that (1) larger shelters host a larger number of fish species and that (2) shelters of a similar size host more fish species when they are clumped together as compared to when they are spread apart.

Using the collected data, we predicted that we could determine species-specific preferences for the dominant species in the study area. We addressed the statistical challenges

associated with low-volume data by employing a unique four-step analysis in our design. This analysis involved the crossed preferences of functional (diet) and systematic (family) groups and their combination, allowing us to determine species-specific preferences. This methodology may offer a significant advantage and may be used to unveil species-specific preferences globally, irrespective of the shelter design.

## 2. Materials and Methods

### 2.1. Location and Experimental Design of the Shelters

As mentioned earlier, to comprehend the preferences of reef fishes, different shelter designs with various characteristics can be employed based on the specific preferences to be explored, especially for each study site. To investigate if a particular feature is advantageous, individuals in the community should be given a choice among several options. Our study was conducted at Katza Beach, Gulf of Eilat, Israel (29°31′24.9″ N, 34°56′09.1″ E). The site was an abandoned oil terminal with support columns anchored to the seabed. We installed balcony-like structures around the columns furthest from the shore (Figure 1). The column diameters were approximately 1 m, and the balcony diameter is 3 m. Each balcony was constructed of galvanized steel with a white epoxy paint coating. The balconies were made of two metal rings, 1 m apart, with a metal mesh between them. These provided high connectivity between the shelters, mirroring a common feature in natural reefs. This connectivity was a significant factor in the distribution, abundance and richness of reef fish [24,34,35,38].

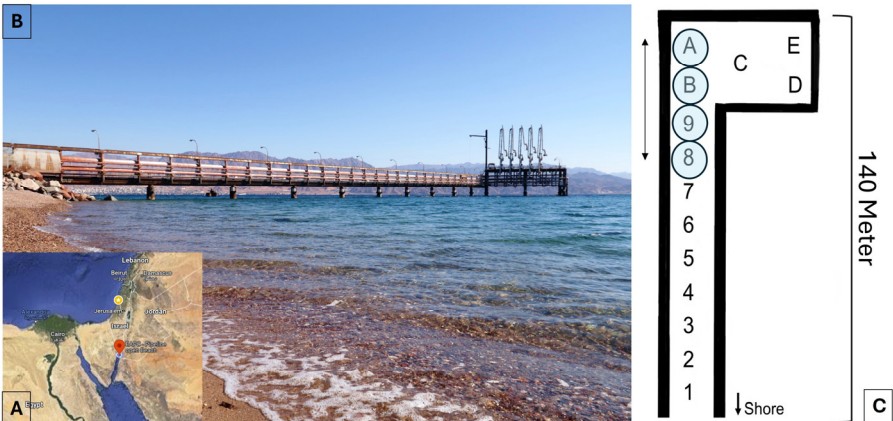

**Figure 1.** The Katza abandoned oil jetty study site. (**A**) Location Google Earth, (**B**) viewed from the south (Picture TS), (**C**) A sketch depicting the columns of the jetty; the columns included in our study are denoted by circles. Columns numbers and letters designation were made during their construction and are marked on them.

The maximum depth around the columns was about 16 m, and the balconies were positioned at 8 to 12.5 m. Column A is approximately 140 m from the shore.

### 2.1.1. Experimental Shelters Design

Small round artificial shelters (RAS) were designed and strategically placed on an abandoned oil jetty at selected locations. The monitoring of the shelters was conducted over a period of ten months. Our design of the shelters aimed to match the size of the most abundant species on the site, which was identified as the Pomacentridae family during preliminary observations. The typical size of Pomacentridae fish ranges from 4.5 to 45 cm in length [39], ensuring that individuals could access the shelters comfortably. We also wanted to keep the shelters in the same scale as the corals in the area and, thus, chose a simple, ball-like shape. All shelters were designed with a relatively high number of holes, as research suggests that reef fish prefer shelters with numerous openings [24,27,29,33].

As mentioned, we centered on two commonly studied characteristics of reef fish shelters: size and spatial distribution. This was achieved by investigating fish preferences

among three different-sized shelters and various spatial distributions (clumped or dispersed) of shelters. To accomplish this, we installed 68 round clay artificial shelters (RAS) with three different-sized volumes—large (10,306 cm$^3$, 27 cm diameter), medium (3315 cm$^3$, 18.5 cm diameter), and small (1437 cm$^3$, 14 cm diameter). The large RAS had an average of 145 holes, the medium 91 holes, and the small 65 holes. Entry-hole diameters were similar in all RAS. These diverse RAS were affixed to the underside of eight balconies on columns 8, 9, A, and B (Figure 1). The balconies were situated at an average depth of 10.2 m (+1.4 SD, range 8–12.5; Appendix A).

### 2.1.2. Experimental Configuration for Dispersed vs. Clumped RAS

We conducted a comparison of reef fish abundance between clumped and dispersed RAS. The design included 32 medium-sized RAS strategically placed around columns 8, 9, and B (Appendix A). We implemented two arrangements on each balcony: (a) four clumped RAS and (b) four dispersed RAS, totaling eight RAS on each balcony. The clumped array was positioned halfway across the balcony area, with the RAS arranged in a cross shape in the middle. In contrast, the dispersed array was placed on the other half of the balcony, positioning the RAS as far apart as possible (Figure 2B–D).

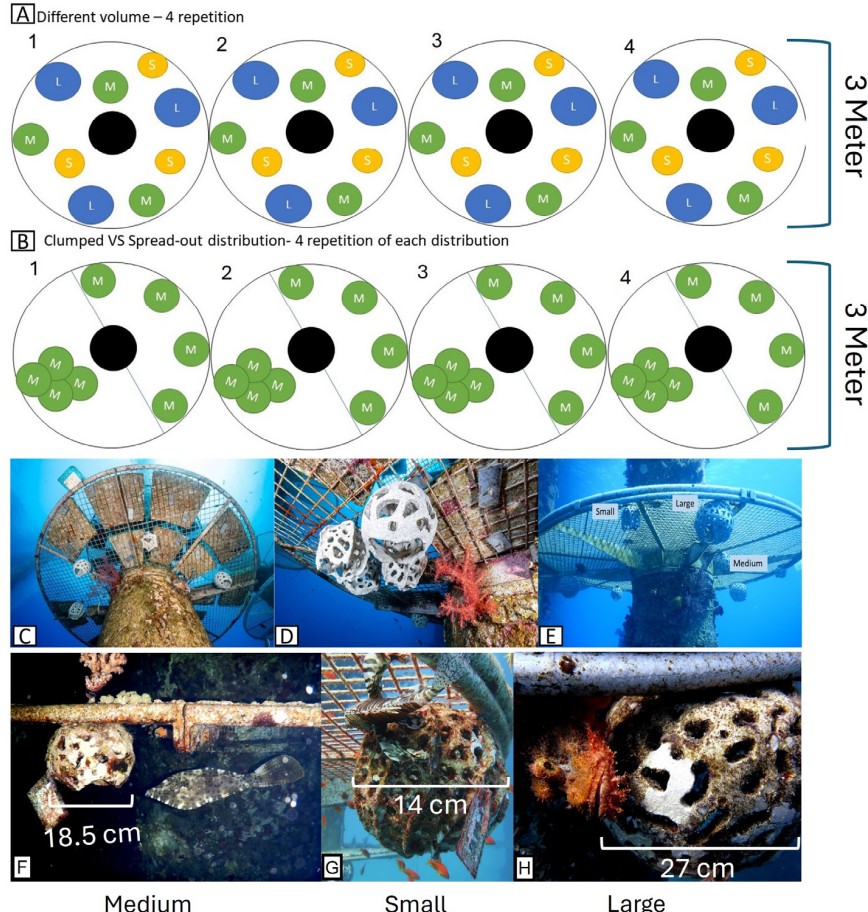

**Figure 2.** Round artificial shelters (RAS) arrangement on the experimental platforms. (**A**)—upper panel: distribution of the different volume RAS—S (small), M (medium), L (large), a total of four balconies and four replicates. (**B**)—middle panel: clumped vs. dispersed distribution, a total of four balconies and four replicates. (**C**)—in situ distribution experiment dispersed arrangement (Picture Boaz Samorai). (**D**)—in situ distribution experiment—clumped arrangement (Picture Boaz Samorai). (**E**)—in situ positioning of large-, medium-, and small-sized RAS (Picture TS). (**F**)—in situ medium-size RAS (Picture TS). (**G**)—in situ small-size RAS. Picture by TS. (**H**)—in situ large-size RAS (Picture TS).

### 2.1.3. Experimental Configuration for the Different Sized RAS

We conducted a comparison of reef fish abundance between clumped and dispersed RAS. The design included 36 RAS, with 12 small, 12 medium, and 12 large (Figure 2). The RAS were placed on the balconies of columns A and B (Appendix A). We placed nine RAS on each balcony, three of each size, as far apart as possible (Figure 2A,D).

### 2.2. Monitoring Procedure and Sampled Areas

The monitoring procedure was consistent for both experiments. The fish communities on the RASs and balconies underwent weekly monitoring for three months, every two weeks for the subsequent two months, once a month for the final three months, and again three months later, resulting in a total observation period of 11 months. Night observations were conducted once a month for nine months. The monitoring frequency was determined by feeding the collected data into an accumulation curve, indicating whether all fish present in the study site were documented [42].

The sampling area for the balconies was defined as the balcony boundary and the circumference of half a meter around it, resulting in a volume of 2.92 m$^3$ for the observation area. For each RAS, the sampling area was defined as the RAS itself and the radius of the RAS around it. The total observation volume for each RAS was 8244 cm$^3$ for the large RAS, 26,522 cm$^3$ for the medium, and 11,494 cm$^3$ for the small RAS. Determining the sampling areas for balconies and RAS was based on preliminary surveys, aiming to include all fish inhibiting the sampling area, as the balconies and RAS were observed to function as attraction sites, with individuals moving around them [24,38]. Sampling included data on species abundance and diversity, and all sampling sessions were conducted using scuba diving.

Survey Technique

Two divers conducted all surveys, with TS as the lead diver and a dive buddy. The observations were documented by writing on a slate and capturing still images with one of two cameras (Panasonic LUMIX DC-FT7, Nikon Coolpix W300; both cameras purchased in Eilat, Israel). Every fish passing through or residing in the study area was meticulously documented. Identification was conducted at the species level for all encountered fish. Certain species, challenging to distinguish, were grouped for reporting purposes: *Acanthurus nigrofuscus* and *Ctenochaetus striatus*, *Caesio lunaris* and *Caesio suevica*, *Kyphosus cinerascens* and *Kyphosus vaigiensis*, *Parupeneus forsskali* and *Parupeneus macronema*, and *Corythoichthys flavofasciatus* and *Corythoichthys schultzi*. Each survey consisted of two parts: the first involved the surveillance of the balcony, and the second focused on the shelters.

Balcony surveillance—Each survey commenced with the divers hovering approximately two meters from the balcony, positioned opposite each other, and moving in opposite directions to gain an optimal total view of the observed area. Both divers documented all visible fish from this distance to minimize disturbance to the fish present. As the survey progressed, divers approached the balconies to identify and document smaller fish (Gobiidae and Blenniidae). The total duration of each observation was four minutes, as preliminary surveys at the site indicated that longer observation times during the daytime resulted in the resampling of the same individuals.

During the night, most of the fish positioned themselves on the balconies or in the RAS for extended periods and were not actively swimming or moving around during the surveys. In cases where there were discrepancies in the number of fish recorded by the two divers, the lead diver's assessment was used for the final record. Fish identification, abundance, documentation, and fish records were then confirmed using images captured during the survey.

RAS monitoring—The RAS monitoring was conducted through each shelter's entrance. Given their small size, the lead diver solely monitored the RAS. Observations of the RAS were not time-based, as preliminary surveys indicated that individuals would not approach

the RAS when being actively observed. Additionally, we observed that individuals close to the RAS would often enter the shelter as the diver approached. The monitoring process involved an initial observation by hovering near the balcony and then circling it twice in a specific order through all the RAS. This was carried out to minimize disturbance to the fish. The first loop was performed approximately one meter from the balcony during day dives and 0.6 m during night dives. During the first loop, the diver documented from the outside all the fish that passed through or inhabited the RAS. The second loop was performed in the same order, with the lead diver swimming as close to the shelters as possible to observe them through the shelter entrance [43,44].

### 2.3. Data Analyses to Categorize the Different Species

Initially, we created an accumulation curve to evaluate species saturation during day and night, ensuring our ability to capture all fish species at the study site. Additionally, statistical tests and manipulations were conducted to ascertain whether there existed a significant difference in fish preferences for the various-sized RAS and their distributions. Unfortunately, a parametric test could not be applied to compare the abundance of different species due to the small number of individuals entering shelters (average {all species included}/shelter/survey N = 1.45 day, N = 0.74 night).

To articulate the specific occurrence of each fish species, we grouped different species into two divisions—systematically by family and functionally by diet. This grouping approach allowed us to obtain a higher total number of individuals, enabling us to examine group frequencies using nonparametric tests effectively. All analyses and comparisons were conducted using Microsoft Office Excel v2312 spreadsheets and R Statistical Software (v4.2.2; R Core Team 2022, [45]).

### 2.3.1. The Classification of Species into Dietary Classes Was Conducted as Follows

The relative abundance for each species was calculated by dividing the number of individuals of the species recorded in all surveys by the total number of fish recorded in all surveys. Relative abundance was computed from the observations of both experiments, including the control balcony "A-low," as both experiments occurred in the same study area. The calculations were performed separately for day and night, taking into account the nocturnal and diurnal behaviors of certain species.

All species with a relative abundance of >10% (during day or night) were functionally grouped based on their reported diet in the literature [26,46–48]; data from both experiments were considered. The relative contribution of a diet group to the total fish abundance was determined by calculating the number of individuals from each diet group divided by the total number of individuals. The species were categorized into five trophic groups: planktivores (N = 8 species), corallivores (N = 2), herbivores (N = 9), benthivores (N = 14), and piscivores (N = 8), totaling 41 species examined.

### 2.3.2. The Species Were Categorized into Families as Follows

The relative contribution of each family was determined by calculating the number of individuals from each family divided by the total number of individuals in both surveys. The analysis considered the five most abundant families and all species recorded in the surveys were examined for each family. A total of 35 species were examined within the families: Pomacentridae (N = 9 species), Serranidae (N = 7), Labridae (N = 11), Acanthuridae (N = 3), and Scaridae (N = 5). After defining the family and diet group for each species, a table was created to facilitate the study of each species using a four-step analysis, focusing on the dominant species. All tests for the different families were performed using the raw data.

### 2.3.3. The Four-Step Analysis Includes

1.  For each family among the five most abundant families in the surveys, preferences for specific shelters were examined by comparing fish abundance at RAS of different

sizes (using the Wilcoxon rank sum test) and fish abundance at different distributions of RAS (using the Sign test).

2. The first step was repeated for each of the five dietary groups, examining specific shelters groups' preferences. This involved comparing fish abundances across RASs of different sizes (using the Wilcoxon rank sum test) and across different distributions of RASs (using the Sign test).

3. We examined 19 species. The predicted preferences were assigned in a table based on species family and dietary association for each species. The overlaying preferences of both groups (family, diet) were highlighted in the table. For example, the species *Pseudanthias squamipinnis*, belonging to the family Serranidae and classified as a planktivore, was predicted to have the same preferences as other species from the family Serranidae and those classified as planktivores.

4. Due to the small number of individuals entering the shelters, it was not feasible to conduct a parametric statistical analysis to determine the shelter preferences of each species. Therefore, the sum of total fish numbers from all surveys conducted during the 10-month experiment for each shelter size or distribution type was used to test the predictions for each species recorded in the previous steps. The total number of fish for each species was recorded in the table and compared to family and dietary group predictions to determine whether the results were consistent with predicted preferences.

## 3. Results

### 3.1. General

A total of 66 dives were conducted, each lasting approximately one hour. The first experiment examined the presence of fish in different RASs and included 5 preliminary dives (4 day, 1 night), 21 day dives, and 9 night dives. The second experiment, which examined different RAS sizes, consisted of 5 preliminary dives (4 during the day, 1 at night), 18 day dives, and 8 night dives. Over approximately ten months, 92 species from 30 families were recorded from surveys conducted in both experiments (Appendix B). The species accumulation curves suggest that all species present in the area during both day and night were likely recorded in both experiments (Figure 3).

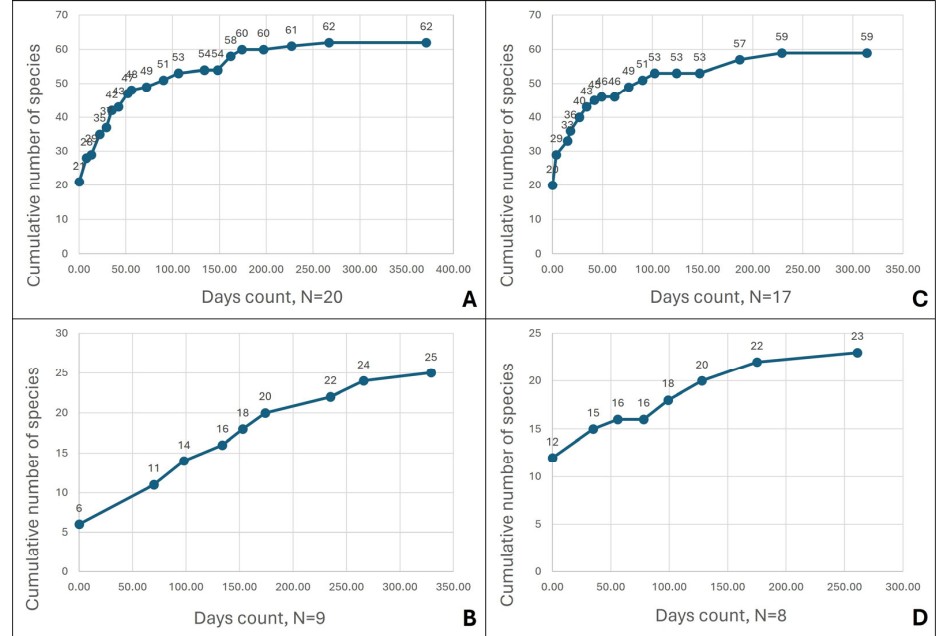

**Figure 3.** Species accumulation curves. Panel (**A**)—day surveys of the experiment with different distributions, (**B**)—night surveys of the experiment with different distributions, (**C**)—day surveys of the experiment with different sizes, (**D**)—night surveys of the experiment with different sizes.

### 3.2. Profiling Species-Specific Preferences

As outlined in the methods section, we combined species based on functional (diet) or systematic (family) traits to uncover the specific preferences of each fish species. We conducted a four-step analysis on each species (refer to Methods), yielding the following results.

#### 3.2.1. Preferences for Shelters across Different Families

We examined the preferences of the most abundant families at the study site: Pomacentridae (40%), Serranidae (35.4%), Labridae (13.1%), Acanthuridae (3.1%), and Scaridae (1.9%). The family Serranidae primarily consists of *Pseudanthias squamipinnis* (99%) (Appendix C).

Pomacentridae—During the day, the medium and large RAS showed higher reef fish abundance values than the small RAS L > S ($p$ = 0.004), M > S ($p$ = 0.056, N = 68). At night, the spread-out dispersal RAS exhibited higher fish numbers than the clumped dispersal ($p$ = 0.043, N = 36).

Serranidae—Throughout the day, the clumped RAS had higher numbers of fish than the spread-out RAS ($p$ = 0.049), N = 80. At night, the clumped RAS displayed higher fish numbers than the spread-out dispersal ($p$ = 0.021, N = 36).

Labridae—During the day, the medium RAS showed higher fish abundance than the small and large RAS M > S ($p$ = 0.0003), M > L ($p$ = 0.006, N = 68). At night, the spread-out dispersal had a higher fish count than the clumped RAS ($p$ = 0.007, N = 36).

Acanthuridae—During the day, the spread-out dispersal had higher fish abundance than the clumped RAS ($p$ = 0.004, N = 80). No fish were observed at night.

Scaridae—During the day, the medium-sized RAS showed a higher fish count than the small RAS; M > S ($p$ = 0.006, N = 68). No fish were observed at night (Table 1).

**Table 1.** Family preferences regarding the size or distribution type of the shelter. "L" stands for large shelters, "M" stands for medium, and "S" for small. "Clumped" stands for clumped dispersal "Dispersed" stands for spread out dispersal, and "NA" means that no significant preferences were found or the data were insufficient to perform statistical tests.

| | Families | | | | | | | | | |
|---|---|---|---|---|---|---|---|---|---|---|
| | Pomacentridae | | Serranidae | | Labridae | | Acanthuridae | | Scaridae | |
| | Day | Night | Day | Night | Day | Night | Day | Night | Day | Night |
| Shelter size preferences | L > S M > S | NA | NA | NA | M > S M > L | NA | NA | NA | M > S | NA |
| Dispersal type preferences | NA | Clumped > Dispersed | Clumped > Dispersed | Clumped > Dispersed | NA | Dispersed > Clumped | Dispersed > Clumped | NA | NA | NA |

#### 3.2.2. Preferences for Shelters across Diet Groups

The percentage of each diet group was calculated by dividing the total number of fish from each group in all surveys by the overall number of individuals recorded. The five dietary groups were as follows: planktivores (N = 8 species, 74%), benthivores (N = 14, 6.6%), herbivores (N = 9, 6%), piscivores (N = 8, 3%), and corallivores (N = 2, ~0%) (Appendix D).

Planktivore—The large RAS exhibited higher fish density values during the day than the small RAS; L > S ($p$ = 0.05, N = 68). No fish were observed at night.

Benthivore—The medium-sized RAS displayed higher fish density during the day than the small RAS; M > S ($p$ = 0.0001, N = 68). At night, the spread-out dispersal configuration had higher fish density than the clumped configuration ($p$ = 0.026, N = 36). No fish were observed at night.

Herbivore—The medium RAS showed higher fish density values during the day than the small RAS M > S ($p$ = 0.005, N = 68). The spread-out dispersal exhibited higher fish density than the clumped dispersal ($p$ = 0.003, N = 80). No fish were observed at night.

Piscivore—The medium-sized RAS displayed higher fish density values during the day than the large RAS; M > L ($p$ = 0.002, N = 68). No fish were observed at night.

Corallivore—The sample size was too small for analysis (Table 2).

**Table 2.** Dietary group preferences regarding shelter size or distribution. "L" stands for large shelters, "M" for medium, and "S" for small RAS. "Clumped" stands for clumped dispersal "Dispersed" stands for spread-out dispersal, "NA" means that no significant preferences were found, or the data was insufficient to perform statistical tests.

| | Diet | | | | | | | | | |
|---|---|---|---|---|---|---|---|---|---|---|
| | Planktivore | | Benthivore | | Herbivore | | Piscivore | | Corallivore | |
| | Day | Night | Day | Night | Day | Night | Day | Night | Day | Night |
| Shelter size preferences | L > S | NA | M > S | NA | M > S | NA | M > L | NA | NA | NA |
| Dispersal type preferences | NA | NA | NA | Dispersed > Clumped | Dispersed > Clumped | NA | NA | NA | NA | NA |

Put as a whole—we could not accept either of our hypotheses as a general rule. It was not proven that our larger shelters host more fish species compared to the smaller shelters, nor that shelters of a similar size host more fish species when they are clumped together compared to when they are dispersed.

### 3.2.3. Using the Two Previous Steps to Profile the Preference of Each Species

For 16 out of 19 species, the results of the family and diet group tests demonstrated an overlap for each species. This implies they exhibited the same preference for size or distribution in the family and the dietary group, indicating distinct preferences. The sample size of the three remaining species (*Aethaloperca rogaa*, *Larabicus quadrilineatus*, *Cephalopholis miniata*) did not permit statistical analysis. Notably, there was no apparent contradiction between species preferences in different categories. No species showed opposing preferences for shelter size or distribution mode between family and dietary group.

As an illustration, consider the species *Pseudocheilinus hexataenia*. We recorded the preferences of its family (Labridae) and dietary group (Benthivore). An overlap between family and dietary group preferences was found during the day and night. Specifically, during the day, there was a preference for medium-sized shelters over small ones, observed in both Labridae and benthivore groups. Similarly, there was a mutual preference for spread-out dispersal over clumped configurations during the night. Additionally, Labridae preferred medium-sized shelters over large ones (Table 3).

**Table 3.** The predicted preferences for the species *Pseudocheilinus hexataenia* align with the preferences identified for the species' family and dietary group, both during the day and night. Here "L" represents large shelters, "M" signifies medium shelters, and "S" indicates small shelters. Additionally, "Clumped" stands for clumped dispersal and "Dispersed" denotes spread-out dispersal.

| *Pseudocheilinus hexataenia* | | | |
|---|---|---|---|
| Labridae | | Benthivore | |
| Day | Night | Day | Night |
| M > S M > L | Dispersed > Clumped | M > S | Dispersed > Clumped |

### 3.2.4. Validation of Predicted Preferences against Total Number of Fish for Each Species

Due to the limited number of individuals entering shelters in both experiments, statistical tests (parametric or non-parametric) for examining species preferences were not feasible. To validate our results, we analyzed the preferences of each species based on the

total number of fish counted in each shelter size or configuration. This count aimed to evaluate the accuracy of species preferences determined by their family and diet.

The total fish count results for the observed species were consistent with the mutual overlap of categories for 12 of 19 species. For the remaining five species, assessments could not be made due to insufficient data, and for two species, no occurrences were recorded. The results included preference predictions for overlapping test scores (crossing the two categories) and partial sub-scores (only one of the categories matched) in each comparison with a minimum of three individuals (N $\geq$ 3, Table 4).

**Table 4.** Shelter preferences for various species based on all conducted surveys, considering each shelter size (large, medium, small) or dispersal pattern (dispersed, clumped). Columns from left to right display the observed species, shelter preferences for each family, preferences for dietary groups, and predicted preferences resulting including both categories. The sums of individuals of the same species counted at each shelter size or dispersal are presented in parentheses. Common results across different classes are highlighted in bold, while partial results from only one of the categories are underlined.

| Species Name | Family | Dietary | Survey Results |
|---|---|---|---|
| *Abudefduf vaigiensis* | Pomacentridae | Planktivore | **Large (37)** > Medium (27) > <u>Small (1)</u> |
| Day | **Large > Small** <br> <u>Medium > Small</u> | **Large > Small** | Dispersed (16) > Clumped (9) |
| Night | Dispersed > Clumped | No significant preferences | Medium (5) > Large (3) > Small (0) <br> Clumped (1) > Dispersed (0) |
| *Pseudanthias squamipinnis* | Serranidae | Planktivore | |
| | Clumped > Dispersed | Large > Small | Large (199) > Medium (132) > <u>Small (98)</u> <br> <u>Clumped (38) > Dispersed (13)</u> |
| | Clumped > Dispersed | No significant preferences | <u>Medium (4) > Large (2) > Small (1)</u> <br> <u>Clumped (10) > Dispersed (1)</u> |
| *Acanthurus nigrofuscus/Ctenochaetus striatus* | Acanthuridae | Herbivore | |
| Day | **Dispersed > Clumped** | **Dispersed > Clumped** <br> Medium > Small | Large (9) > Medium (7) > Small (6) <br> **Dispersed (36) > Clumped (14)** |
| Night | **No presence** | **No presence** | **No presence** |
| *Bodianus anthioides* | Labridae | Benthivore | |
| Day | **Medium > Small** <br> Medium > Large | **Medium > Small** | Large (2) > Medium (0) = Small (0) <br> Dispersed (3) = Clumped (3) |
| Night | **Dispersed > Clumped** | **Dispersed > Clumped** | Large (0) = Medium (0) = Small (0) <br> **Dispersed (19) > Clumped (7)** |
| *Oxycheilinus mentalis* | Labridae | Piscivore | |
| Day | <u>Medium > Small</u> <br> **Medium > Large** | **Medium > Large** | **Medium (10)** > <u>Small (5)</u> > **Large (1)** <br> Dispersed (13) < <u>Clumped (6)</u> |
| Night | Dispersed > Clumped | No significant presence | Large (0) = Medium (0) = Small (0) <br> Dispersed (3) > Clumped (2) |
| *Pomacentrus trichrourus* | Pomacentridae | Planktivore | |
| Day | Medium > Small <br> **Large > Small** | **Large > Small** | Medium (1) > Large (0) = Small (0) <br> Clumped (6) > Dispersed (0) |
| Night | Dispersed > Clumped | No significant preferences | Large (2) > Medium (1) > Small (0) <br> Clumped (1) = Dispersed (1) |
| *Thalassoma lunare* | Labridae | Benthivore | |
| Day | **Medium > Small** <br> Medium > Large | **Medium > Small** | **Medium (19)** > Large (17) > **Small (8)** <br> Dispersed (8) < Clumped (5) |
| Night | **Dispersed > Clumped** | **Dispersed > Clumped** | No presence |
| *Neopomacentrus miryae* | Pomacentridae | Planktivore | |
| Day | Medium > Small <br> **Large > Small** | **Large > Small** | **Large (47)** > Medium (29) = **Small (22)** <br> <u>Dispersed (8) > Clumped (1)</u> |
| Night | <u>Dispersed > Clumped</u> | No significant preferences | Medium (67) > Large (64) = Small (59) <br> <u>Dispersed (65) > Clumped (38)</u> |
| *Thalassoma rueppellii* | Labridae | Benthivore | |
| Day | **Medium > Small** <br> Medium > Large | **Medium > Small** | **Medium (56)** > Large (37) > **Small (28)** <br> Dispersed (36) < <u>Clumped (24)</u> |
| Night | **Dispersed > Clumped** | **Dispersed > Clumped** | No presence |
| *Scarus ferrugineus* | Scaridae | Herbivore | |
| Day | **Medium > Small** | <u>Dispersed > Clumped</u> <br> **Medium > Small** | **Medium (8)** = Large (8) > **Small (1)** <br> Dispersed (6) < Clumped (1) |
| Night | **No presence** | **No presence** | **No presence** |

**Table 4.** *Cont.*

| Species Name | Family | Dietary | Survey Results |
|---|---|---|---|
| *Zebrasoma xanthurum* | Acanthuridae | Herbivore | |
| Day | **Dispersed > Clumped** | **Dispersed > Clumped**<br>Medium > Small | Small (1) > Large (0) = Medium (0)<br>Clumped (1) > Dispersed (0) |
| Night | **No presence** | **No presence** | **No presence** |
| *Cephalopholis miniata* | Serranidae | Piscivore | |
| Day | Clumped > Dispersed | Medium > Large | No presence |
| Night | Clumped > Dispersed | No significant presence | No presence |
| *Pseudocheilinus Hexataenia* | Labridae | Benthivore | |
| Day | **Medium > Small**<br>Medium > Large | *Day*<br>**Medium > Small** | Large (1) > Medium (0) = Small (0)<br>Dispersed (0) = Clumped (0) |
| Night | **Dispersed > Clumped** | **Dispersed > Clumped** | No presence |
| *Amblyglyphidodon indicus* | Pomacentridae | Planktivore | |
| Day | Medium > Small<br>**Large > Small** | **Large > Small** | Large (0) = Medium (0) = Small (0)<br>Dispersed (13) > Clumped (7) |
| Night | Dispersed > Clumped | No significant preferences | Large (1) > Medium (0) = Small (0)<br>Clumped (2) > Dispersed (1) |
| *Bodianus diana* | Labridae | Benthivore | |
| Day | **Medium > Small**<br>Medium > Large | **Medium > Small** | Large (1) > Medium (0) = Small (0)<br>Dispersed (2) > Clumped (1) |
| Night | **Dispersed > Clumped** | **Dispersed > Clumped** | No presence |
| *Aethaloperca rogaa* | Serranidae | Piscivore | |
| Day | Clumped > Dispersed | Medium > Large | Small (1) > Large (0) = Medium (0)<br>Clumped (1) > Dispersed (0) |
| Night | Clumped > Dispersed | No significant presence | No presence |
| *Scarus niger* | Scaridae | Herbivore | |
| Day | **Medium > Small** | Dispersed > Clumped<br>**Medium > Small** | Medium (1) = Large (1) > Small (0)<br>Dispersed (0) = Clumped (0) |
| Night | **No presence** | **No presence** | **No presence** |
| *Calotomus viridescens* | Scaridae | Herbivore | |
| Day | **Medium > Small** | Dispersed > Clumped<br>**Medium > Small** | **Medium (7) > Small (3)** > Large (0)<br>Dispersed (3) > Clumped (2) |
| Night | **No presence** | **No presence** | **No presence** |
| *Larabicus quadrilineatus* | Labridae | Corallivore | |
| Day | Medium > Small<br>Medium > Large | Not enough data | No presence |
| Night | Dispersed > Clumped | Not enough data | No presence |

The correspondence between predictions and fish counts at the study site suggests that the predictions, derived from our unique four-step analysis, are reliable and accurately represent the actual preferences of the observed reef fish in the study site.

## 4. Discussion

Traditional conservation measures, such as no take-zones, nature reserves, and marine protected areas, often need to be revised to achieve conservation goals amid the ongoing deterioration of coral reefs [9,16]. This has led to a growing emphasis on 'active' restoration [6,9,14] involving the deployment of artificial shelters (AR). These structures aim to offer reef fish the necessary protection from predators or human disturbances while supporting essential processes for their survival [5,27–29,35,38,49,50]. Indeed, ARs may deliver similar functions and benefits to the marine environment as natural reefs, including increased fish abundance, enhanced coral cover, preservation of target species, and more [16].

Despite the increasing popularity of AR use [12–15], the varied designs and methodologies employed in different studies [15,18,23,26,28,36] highlight the lack of consensus on the most effective approach and characteristic needed for constructing successful AR to achieve various conservation goals, such as enhancing reef fish abundance or richness in a given community. Assessing the "success" of specific shelter designs across communities may not be universally applicable, given the diverse physical parameters and community structures at play.

Moreover, studying fish preferences at the community level raises uncertainties, as the preferences of specific species often get obscured within the collective preferences of differ-

ent species. Research has demonstrated that no one-size-fits-all shelter can accommodate all fish species. Each species requires a distinct habitat characteristic [36,37], influencing the composition of the fish community seeking refuge in the shelters. Therefore, tailoring shelter designs to benefit specific species is crucial for the success of each conservation initiative. For instance, some restoration efforts may prioritize increasing the abundance of key species that have declined due to invasive species. In contrast, others may aim to accommodate species highly adapted to specific corals, the abundance of which has decreased due to factors like storms, diseases, or bleaching. In each case, a targeted shelter design is essential to attract and support the needs of the intended species. Establishing design preferences for shelters should precede widespread implementation [11], and to achieve that, uncovering specific species preferences becomes imperative.

To unveil the individual preferences of each species, we initiated our study by examining the community structure and species composition at our study site. Focusing on two commonly studied characteristics, different sizes and spatial distribution of the shelters, we designed and deployed shelters of varying sizes and dispersal types [5,27–30,34,35]. These shelters were affixed to an abandoned oil jetty in Katza Beach, Eilat, Israel. Opting for small shelters was crucial, as they closely mimic the shelters available to fish in natural reefs, making them essential for our study. Over 11 months, we conducted fish surveys during the day and night to identify successful shelters based on the abundance of individuals in the area. Our collected data, and species accumulation curves (Figure 3), revealed the comprehensive representation of all species present in the study area throughout the surveys.

The functional diet grouping of the species yielded results consistent with those in the Gulf of Aqaba [25,51]. Notably, Pomacentridae (40%), Serranidae (35%), and Labridae (13%) emerged as the most abundant families. These findings align with studies conducted in the Gulf of Aqaba over both short (five months, [51]) and long (six years, [25]) durations. This supports the notion proposed by Higgins [16] that a monitoring period ranging from a few months to a few years can effectively track fish populations in ARs, given the relatively short life expectancy (under five years) of many fish species. For instance, Pomacentrid species, known for their high site fidelity and small territories or home ranges [52], suggest that successful shelters could provide a viable long-term solution for this community.

Our next objective was to uncover the shelter preferences of the dominant species based on their relative abundance and frequency. However, we encountered two significant challenges. First, tracking the individual preferences of each species in the study area proved challenging, demanding substantial time and effort. Second, examining small shelters presented a limitation for statistical analysis due to frequently low fish numbers. Consequently, we could not conduct statistical tests using only the individuals of a single species. We addressed both challenges by forming groups that share a joint base regarding taxonomy and diet. Grouping multiple species allowed for a sufficient fish count for statistical tests, enabling us to determine each group's preferences concerning shelter sizes and distributions. We were glad to discover that both classifications provided clear and meaningful preferences for different RAS sizes and distributions. However—this came with the cost of disregarding other traits, behaviors, and life histories. Other researchers may choose their grouping approaches.

We profiled each fish species using both group classifications to extrapolate individual preferences from group preferences. We compared the results of the relevant for each species, considering both family and diet type. This analytical approach yielded a unique preference profile for each species. In some instances, the same preference for shelter size or dispersal mode was consistent in both groups of the same species. For example, *Oxycheilinus mentalis* exhibited a preference for medium-sized shelters during the day in both the family group (Labridae) and the dietary group (piscivores). Similarly, *Acanthurus nigrofuscus* and *Ctenochaetus striatus* preferred dispersed distribution during the day in both the family (Acanthuridae) and dietary group (herbivores).

When examining our original hypotheses, we find them somewhat naïve. Indeed, overall we could not accept either of our hypotheses as a general rule. It was not proven that larger shelters host more fish species compared to smaller shelters, nor that shelters of a similar size host more fish species when they are clumped together compared to when they are dispersed. The shelter preferences were not general across species of functional groups. While some species preferences differed between the two classifications, the groups were never contradictory. For instance, *Thalassoma lunare* was found to prefer medium-sized shelters over large shelters during the day according to the family (Labridae) group, a preference not observed in its functional group (benthivores). However, when examining both classifications, the species preferred medium-sized shelters over small shelters, supported by both family (Labridae) and dietary groups (benthivores). Importantly, the diverse preferences extracted for each species from the systematic and functional groups did not contradict each other (Table 4).

To verify the accuracy and the alignment of the preferences we identified with the actual presence of the fish, on-site surveys were conducted. Due to the impracticality of statistical tests on individual numbers for certain species, we aggregated the abundance of each fish species across all surveys for each shelter size or dispersal type. The results of the shelters' fish counts concurred with the statistical tests outcomes in both classifications. Some results were consistent across both classifications, while others were partial, aligning with only one classification. Notably, the results were deemed accurate when the difference in fish numbers between comparisons exceeded two individuals (Table 4).

The validation of predicted preferences through on-site fish counts underscores our four-step analysis's high value and significance. This validation also supports our prediction, that profiling species-specific preferences for shelters is possible even when using low-volume data by using grouping analysis. The challenge of determining statistical significance, particularly with low-volume data, is a common issue in science. Overcoming this obstacle validates our findings and also opens avenues for future research on specialized artificial shelters that can significantly benefit the marine environment.

Moreover, using a few shelters within a relatively short time frame offers several advantages. It makes research more efficient, less time-consuming, and more cost-effective. Employing a limited number of shelters for studying fish populations before widespread implementation allows us to tailor optimal shelters to the existing fish community, considering our conservation goals while minimizing disturbance to the marine environment.

It is important to note that our shelter design was specifically chosen to accommodate small reef fish abundant at the site and serve as a research tool. We do not recommend mass construction of such RASs to be used in reef restoration, but instead, finding the target fishes' specific needs and catering to them. Furthermore, this design may not be suitable for fisheries or for accommodating large, non-site-attached fish. We observed that individual fish and species exhibited distinct and specific shelter choices [24,28,50]. These variations likely stem from a combination of factors, including individual size, shelter size, diet, and individual preferences [5,27–32,35,53,54]. Notably, each species displayed unique shelter size preferences and we could not identify a universal pattern applicable to all species.

Furthermore, our findings indicate that species shelter preferences at night differ from those during the day. For instance, shelters during the day may offer protection from diurnal predators, solar radiation, or water currents, concerns that might not be as relevant at night [55]. Studying the nocturnal behavior of reef fish is more complex, primarily due to the challenges posed by limited visibility.

During daylight hours, only 4 of the 17 species preferred large shelters over medium and small ones and a preference for medium over small shelters (large > medium > small). Conversely, 6 out of 17 species favored medium-sized shelters over large and small ones. Interestingly, at night, individuals of the species *Neopomacentrus miryae* were observed to prefer medium-sized shelters over large and small ones (medium > large > small).

The significant differences in fish numbers observed in various shelters during day and night suggest that the placement and distribution of individuals are not random.

Instead, it appears that individuals of different species make deliberate choices, exhibiting preferences for specific features of different shelters [24,28,50]. An illustrative example involves *Neopomacentrus miryae*, family Pomacentridae, a planktivore fish active during the day, forming schools of several hundred individuals during feeding [51]. Interestingly, our observations suggest that they prefer shelters when they are not actively feeding but seeking a resting place at night.

During our day surveys, *N. miryae* formed large schools of 200–300 individuals near the balconies. We expected a similar grouping at night, but to our surprise, the schools dispersed, and most surveys revealed an average of only two individuals in each shelter, regardless of its size. Given the limited number of shelters, 36 in this experiment, indicate a deliberate choice to disperse and seek more distant shelters, avoiding clustering. Importantly, the physical size of the shelters did not limit individuals, as small shelters could accommodate 20–25 individuals of the species.

Additionally, we noted a pattern in *N. miryae*'s behavior over time. In the initial weeks, the average fish count was around one individual per shelter. After several weeks, the density increased to approximately four individuals in the same shelter. Over the subsequent months, the observed density decreased to one or two individuals per shelter, remaining consistent for the experiment. This behavior was exhibited by both small (juveniles) and large individuals throughout the study period, indicating that size did not influence the observed densities. The deliberation decision by these fish to avoid aggregation, even at the cost of actively seeking more distant shelters, exemplifies the intricate decision-making process individuals undergo when selecting a shelter for the night (Figure 4).

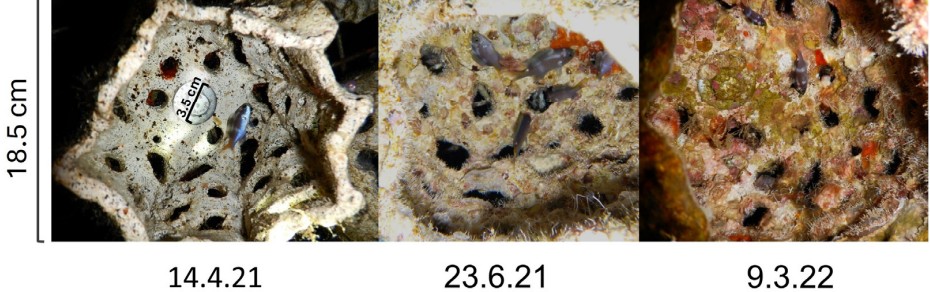

**Figure 4.** *Neopomacentrus miryae* fish in medium-sized shelters at night over an 11-month period. Note the accumulation of both fish and of live coverage on the RAS over time.

For certain species, we identified preferences for specific shelter sizes or distribution— *Bodianus anthioides*, which, like *N. miryae*, predominantly used shelters at night. However, unlike *N. miryae*, *B. anthioides* preferred spread-out dispersal over clumped arrangements at night. Remarkably, *B. anthioides* also demonstrated an almost unnatural vertical usage of the shelters (Figure 5).

These findings strongly indicate that specific species favored certain shelters. In line with species-sorting theory, the occurrence of a species in a particular place is influenced by the favorability of the environment, which can be biotic or abiotic. Species sorting can lead to distinct niches, each housing one or more species [56]. To prevent the separation of species in the design of artificial shelters, it becomes crucial to consider the heterogeneity of these shelters. Distributing different shelters throughout the habitat or study area becomes essential to achieve optimal diversity and accommodate the varying preferences of different species.

While specific shapes, sizes, or designs may be more advantageous for some species, the same might not be the case for others [36,37]. For that reason, we encourage examining various shelter designs to find the best fit for the target species. Our approach is straightforward: identify the preferred shelter for the target species (such as endangered or key species). Shelters with varying characteristics, shelter size, or spatial distribution

should be positioned in the study site and monitored. The data recorded regarding species choice among the different shelters can then be used in our four-step analysis to discover the optimal shelter for the specific desired goal.

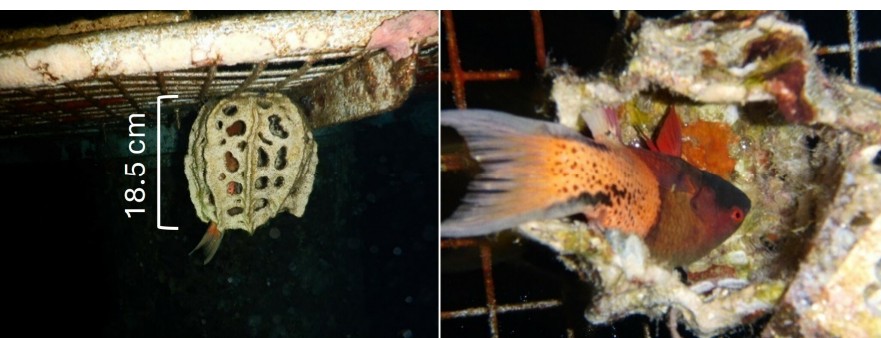

**Figure 5.** A *Bodianus anthioides* inside a shelter from the spread-out dispersal at night. Note that the fish does not fit entirely in the shelter but still tries to hide inside it with the tail sticking out.

## 5. Conclusions

When designing an artificial reef, one must consider a range of constraints such as space availability, changing structural topography, and budget constraints. Furthermore, the human dimension should not be ignored. Research shows divers are likely to choose to dive at sites containing artificial reefs and, by that, reduce the diving pressure and anthropogenic damage to the natural reef [57–60]. Furthermore, the deployment of artificial reefs can add an economical value to the natural features of a diving area [57].

In our study, we strategically placed artificial shelters on an abandoned oil jetty. Jetties, common submerged structures worldwide, offer opportunities to benefit the marine environment while minimizing natural constraints such as limited space availability and dynamic topography changes. Furthermore, a well-designed artificial reef in an accessible location has the potential to attract divers to the site [40,41]. This underscores the importance of considering both ecological and human-related factors when planning and implementing artificial reef projects.

Our data offers practical insights into prioritizing shelter features within space, budget, and topography constraints. The knowledge gained from this study can guide the development of tailored conservation strategies that consider the specific habitat requirements of various reef fish species. Understanding the preferences of each species enables the design of appropriate shelters, supporting key species or those in decline in the region.

We may uncover further preference patterns by exploring additional subclasses, such as age or group size. This species-specific approach to habitat conservation promises to achieve more significant conservation outcomes, addressing the unique ecological needs of each species within the coral reef ecosystem. Integrating species-specific shelter preferences into conservation and restoration practices can contribute significantly to preserving biodiversity and the ecological functioning of coral reef ecosystems globally.

By aligning goals, defining problems and limitations, and investing time in understanding solutions, we can use artificial shelters more intelligently and purposefully, preventing the waste of limited resources. Adopting this model allows researchers to conclude their specific design in their study site while investing minimal resources, increasing the likelihood of success. This holistic and species-specific approach ensures more effective and sustainable use of artificial shelters in coral reef conservation efforts.

**Author Contributions:** Methodology, T.S., Y.Z., R.Y. and N.S.; Validation, R.Y. and N.S.; Formal analysis, Y.Z. and N.S.; Investigation, T.S.; Resources, N.S.; Data curation, T.S.; Writing—original draft, T.S. and R.Y.; Writing—review and editing, T.S., Y.Z., R.Y. and N.S.; Supervision, Y.Z., R.Y. and N.S.; Funding acquisition, N.S. All authors have read and agreed to the published version of the manuscript.

**Funding:** This research received no external funding.

**Institutional Review Board Statement:** Not applicable.

**Informed Consent Statement:** Not applicable.

**Data Availability Statement:** Data are contained within the article.

**Acknowledgments:** We are grateful for the assistance of Natalie Chernihovsky, Roi Holzman, Gil Rilov, and Daniel Golani in fish identification and in focusing on getting into the minds of the fish. Rilov was especially helpful in sharing his expertise regarding the study site. We also thank the many individuals who gladly helped with the data collection in the field—Reem Neri, Inbal Kahan, Omer Waizman, Lior Benzer, Ron Jano, Lisa Schmidt, Keren Or Rinkov, Natalie Chernihovsky, Neil Brosh, Saurav Dutta, Roi Feinstein, Tom Leu, Tomer Ketner, Asa Oren, Dor Shefy, Clara Seinsche, Almog Ben-Natan, Josey Cory Wright, Kerem Çıtak, Inbal Carmel, and Ian Segal.

**Conflicts of Interest:** The authors declare no conflicts of interest.

## Appendix A

RAS distribution on the different balconies and poles. Column designation was set during jetty construction—see Figure 1. "Experiment" with the experiment type: "Dispersed vs. clumped" refers to the dispersed vs. clumped distributions; and "Different sized RAS" represents the observation of three different sized shelters.

| Column Number | Balcony Depths (Meters) | Experiment |
|:---:|:---:|:---:|
| 8 | 9.5 | Dispersed vs. clumped |
| 9 | 10 | Dispersed vs. clumped |
|  | 11.4 | Dispersed vs. clumped |
| A | 8.7 | Different sized RAS |
|  | 10.1 | Different sized RAS |
|  | 11.3 | Control |
| B | 8 | Different sized RAS |
|  | 10.6 | Different sized RAS |
|  | 12.5 | Dispersed vs. clumped |

## Appendix B

Scientific names of each species and family recorded during the surveys. Total of 92 species from 30 different families.

**Acanthuridae**
1. *Acanthurus nigrofuscus/Ctenochaetus striatus*
2. *Zebrasoma desjardinii*
3. *Zebrasoma xanthurum*
**Antennariidae**
4. *Antennatus coccineus*
**Apogonidae**
5. *Apogon erythrosoma*
6. *Cheilodipterus novemstriatus*
7. *Ostorhinchus cyanosoma*
**Balistidae**
8. *Sufflamen albicaudatus*
**Blenniidae**
9. *Ecsenius dentex*
10. *Ecsenius frontalis*
11. *Ecsenius gravieri*
12. *Ecsenius midas*
13. *Mimoblennius cirrosus*
14. *Plagiotremus tapeinosoma*
15. *Blenniidae Sp.*
**Caesionidae (Fusiliers)**
16. *Caesio lunaris/Caesio suevica*
**Chaetodontidae (Butterflyfishes)**
17. *Chaetodon auriga*
18. *Chaetodon austriacus*
19. *Chaetodon fasciatus*
20. *Heniochus intermedius*
21. *Chaetodon paucifasciatus*

22. *Chaetodon trifascialis*
**Congridae**
23. *Conger cinereus*
**Gobiidae**
24. *Eviota guttata*
25. *Gobiidae familly (Unknown)*
26. *Pleurosicya micheli*
**Holocentridae**
27. *Myripristis murdjan*
28. *Neoniphon sammara*
29. *Sargocentron diadema*
30. *Sargocentron rubrum*
**Kyphosidae**
31. *Kyphosus cinerascens/Kyphosus vaigiensis*
**Labridae (Wrasses)**
32. *Bodianus anthioides*
33. *Bodianus diana*
34. *Cheilinus abudjubbe*
35. *Cheilinus lunulatus*
36. *Coris aygula*
37. *Labroides dimidiatus*
38. *Larabicus quadrilineatus*
39. *Oxycheilinus mentalis*
40. *Pseudocheilinus hexataenia*
41. *Thalassoma lunare*
42. *Thalassoma rueppellii*
**Lethrinidae**
43. *Monotaxis grandoculis*
**Monacanthidae**
44. *Aluterus scriptus*
45. *Cantherhines pardalis*
**Mullidae**
46. *Parupeneus cyclostomus*
47. *Parupeneus forsskali/Parupeneus macronema*
**Muraenidae**
48. *Gymnothorax flavimarginatus*
49. *Gymnothorax griseus*
50. *Gymnothorax pharaoensis*
51. *Muraenidae sp.*
**Ostraciidae**
52. *Ostracion cubicus*
**Pempheridae**
53. *Pempheris vanicolensis*
**Pomacanthidae (Angelfishes)**
54. *Apolemichthys xanthotis*
55. *Pomacanthus imperator*
**Pomacentridae (Damselfishes)**
56. *Abudefduf vaigiensis*
57. *Amblyglyphidodon flavilatus*
58. *Amblyglyphidodon indicus*
59. *Chromis viridis*
60. *Chromis weberi*
61. *Dascyllus trimaculatus*
62. *Marginate dascyllus*
63. *Neopomacentrus miryae*
64. *Pomacentrus trichrourus*
**Pseudochromidae (Dottybacks)**
65. *Pseudochromis fridmani*
66. *Pseudochromis olivaceus*
67. *Pseudochromis springeri*
**Scaridae (Parrotfishes)**
68. *Calotomus viridescens*
69. *Chlorurus gibbus*
70. *Chlorurus sordidus*
71. *Scarus ferrugineus*
72. *Scarus niger*
**Scorpaenidae**
73. *Pterois miles*
74. *Scorpaenodes corallinus*
75. *Scorpaenopsis oxycephala*
**Serranidae**
76. *Aethaloperca rogaa*
77. *Cephalopholis argus*
78. *Cephalopholis hemistiktos*
79. *Cephalopholis miniata*
80. *Epinephelus fasciatus*
81. *Pseudanthias squamipinnis*
82. *Pseudanthias taeniatus*
**Siganidae**
83. *Siganus argenteus*

84. *Siganus rivulatus*
85. *Siganus stellatus*
**Sparidae**
86. *Diplodus noct*
**Syngnathidae**
87. *Corythoichthys flavofasciatus/Corythoichthys schultzi*
**Synodontidae**
88. *Synodus variegatus*
**Tetraodontidae**
89. *Arothron diadematus*
90. *Arothron hispidus*
91. *Canthigaster margaritata*
**Tripterygiidae**
92. *Norfolkia brachylepis*

## Appendix C

The five most abundant families and the species recorded for each family.

| Pomacentridae (N = 9) | Serranidae (N = 7) | Labridae (N = 11) | Acanthuridae (N = 3) | Scaridae (N = 5) |
|---|---|---|---|---|
| 1. *Abudefduf vaigiensis* 2. *Dascyllus trimaculatus* 3. *Marginate dascyllus* 4. *Chromis weberi* 5. *Neopomacentrus miryae* 6. *Amblyglyphidodon indicus* 7. *Amblyglyphidodon flavilatus* 8. *Pomacentrus trichrourus* 9. *Chromis viridis* | 1. *Pseudanthias squamipinnis* 2. *Cephalopholis miniata* 3. *Cephalopholis argus* 4. *Pseudanthias taeniatus* 5. *Aethaloperca rogaa* 6. *Epinephelus fasciatus* 7. *Cephalopholis hemistiktos* | 1. *Bodianus anthioides* 2. *Bodianus diana* 3. *Cheilinus abudjubbe* 4. *Cheilinus lunulatus* 5. *Coris aygula* 6. *Labroides dimidiatus* 7. *Larabicus quadrilineatus* 8. *Oxycheilinus mentalis* 9. *Pseudocheilinus hexataenia* 10. *Thalassoma lunare* 11. *Thalassoma rueppellii* | 1. *Acanthurus nigrofuscus/Ctenochaetus striatus* 2. *Zebrasoma desjardinii* 3. *Zebrasoma xanthurum* | 1. *Calotomus viridescens* 2. *Chlorurus gibbus* 3. *Chlorurus sordidus* 4. *Scarus ferrugineus* 5. *Scarus niger* |

## Appendix D

The five dietary groups and the species recorded and considered for each group.

| Planktivore (N = 8) | Corallivore (N = 2) | Herbivore (N = 9) | Benthivore (N = 14) | Piscivore (N = 8) |
|---|---|---|---|---|
| 1. *Pseudanthias squamipinnis* 2. *Abudefduf vaigiensis* 3. *Caesio lunaris/Caesio suevica* 4. *Dascyllus trimaculatus* 5. *Chromis weberi* 6. *Neopomacentrus miryae* 7. *Amblyglyphidodon indicus* 8. *Pomacentrus trichrourus* | 1. *Chaetodon paucifasciatus* 2. *Larabicus quadrilineatus* | 1. *Acanthurus nigrofuscus Ctenochaetus striatus* 2. *Scarus ferrugineus* 3. *Ostracion cubicus* 4. *Zebrasoma xanthurum* 5. *Scarus niger* 6. *Ecsenius dentex* 7. *Ecsenius gravieri* 8. *Calotomus viridescens* 9. *Ecsenius frontalis* | 1. *Thalassoma rueppellii* 2. *Thalassoma lunare* 3. *Bodianus anthioides* 4. *Parupeneus forsskali/Parupeneus macronema* 5. *Pseudochromis fridmani* 6. *Corythoichthys flavofasciatus/Corythoichthys schultzi* 7. *Pleurosicya micheli* 8. *Eviota guttata* 9. *Bodianus diana* 10. *Pseudochromis olivaceus* 11. *Neoniphon sammara* 12. *Pseudochromis springeri* 13. *Pseudocheilinus hexataenia* 14. *Gymnothorax pharaoensis* | 1. *Oxycheilinus mentalis* 2. *Pterois miles* 3. *Cephalopholis miniata* 4. *Synodus variegatus* 5. *Cheilodipterus novemstriatus* 6. *Aethaloperca rogaa* 7. *Plagiotremus tapeinosoma* 8. *Scorpaenodes corallinus* |

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
