# Peer review of "Deciphering Preferences for Shelter Volume and Distribution by Coral Reef Fish, Using Systematic and Functional Grouping"

_jmse, doi:10.3390/jmse12010186_

Round 1

Reviewer 1 Report

Comments and Suggestions for Authors

It is an interesting research that deserves attention after some corrections:

The main ones are:

1. This small number of individuals that you sampled justify the investment in coral restoration? The manager will need hundreds or even thousands of shelters, no?

2. It is important to raise a hypothesis that will guide your sampling design, analytical methodology and discussion.

3. I strongly recommend to include some pictures of the different artificial shelters (with scale)

4. recruitment denotes reproduction. this is not the case. here, it is more abundance, colonization, attraction. Why recruitment? there are not small adults?

5. when you group different species you may not consider different behaviors, different life history. 

6.  I recommend to start the discussion with all the limitations of the research and the results and how this could have influence the practical use of these structures in a broad (large) scale. Also, it is important to show how to overcome these limitations, if this is possible.

7.  size scales in all pictures please.

8. It is important to make clear to the readers that this could be apply when the issue is artificial reefs to coral reef fish, to restoration of corral reefs. But when the goal is fishries, this small shelters to dominant species might not function. But I believe that in the first case, it will be a good strategy to reduce human disturbance in this coral reef ecosystem.

Author Response

It is an interesting research that deserves attention after some corrections:

The main ones are:

  1. This small number of individuals that you sampled justify the investment in coral restoration? The manager will need hundreds or even thousands of shelters, no?

We want to thank the reviewer for this comment since it is an indication that we failed to explain the idea behind the methodology we offer here. We are not suggesting the use of our small shelters in restoration but rather the method to find out what fishes want. We added clarification for it in the abstract in lines 27-29:

These profiles may later allow the establishment of ecological-oriented artificial reefs. Moreover, this method can be applied using any design of shelters, size, and research site.”

We also added further clarification in the discussion, lines 512-515:

“It is important to note that our shelter design was specifically chosen to accommodate small reef fish abundant at the site and serve as a research tool. We do not recommending mass construction of such RASs to be used in reef restoration, but rather finding the needs of the target fishes and catering for their needs.

Furthermore, we address the topic at the beginning of the method section, at lines 118-120:

“in order to understand the preferences of reef fishes, different shelter designs with different characteristics can be used depending on the desired preferences to be explored, particularly for each study site “

And we address the topic at the end of the discussion, lines 562-563:

By using this model, researchers can draw conclusions about their own specific design in their own study site by investing minimal resources while increasing their chances of success."

  1. It is important to raise a hypothesis that will guide your sampling design, analytical methodology, and discussion.

- Now included in the final paragraph of the introduction. Lines 95-100 now read

Using the collected data, we hypothesized that we could determine species-specific preferences for the dominant species in the study area. We addressed the statistical challenges associated with low-volume data by employing a unique four-step analysis in our design. This analysis involved the crossed preferences of functional (diet) and systematic (family) groups, and their combination, allowing us to determine species-specific preferences.

  1. I strongly recommend to include some pictures of the different artificial shelters (with scale)

-We added pictures of the artificial shelters to Figure 2 with a scale

  1. recruitment denotes reproduction. this is not the case. here, it is more abundance, colonization, attraction. Why recruitment? there are not small adults?

- Thank you, we agree and have accordingly modified it to abundance.

  1. when you group different species you may not consider different behaviors, different life history. 

We agree and we apologize we failed to explain this point clearly. In our study, we included several parameters to classify (group) the species. These can be expanded by others as needed. However, the point of our study is to show how we can reach conclusions even with minimal sample sizes and classifications, by grouping several species with a common ground. This is important because to date most studies are rejected due to small sample sizes and we show that this is not always correct. Crossing the preferences of the groups reveals a unique profile for a single species. We added further clarification in the discussion at lines 71-472:

However- this came with the cost of possible disregarding other traits, behaviors and life histories. Other researchers may wish to choose their own grouping approaches.

  1. 6.  I recommend to start the discussion with all the limitations of the research and the results and how this could have influence the practical use of these structures in a broad (large) scale. Also, it is important to show how to overcome these limitations, if this is possible.

Thank you for your comment. It seems we failed to make clear that our method is a solution we offer for the problem that arises when trying to profile the preferences of specific species when using low-volume data. We inserted a detailed explanation in the introduction and the discussion in the following lines:

Line 80-83:

However, characterizing the preferences of individual species in the study area can be challenging and requires a significant investment of resources. Additionally, small shelters, common in natural reefs [24,38], often exhibit inherently low fish numbers, posing challenges for statistical tests due to the limited data

Lines 461-469:

However, we encountered two significant challenges. Firstly, tracking the individual preferences of each species in the study area proved challenging, demanding substantial time and effort. Secondly, the examination of small shelters presented a limitation for statistical analysis due to frequently low fish numbers. Consequently, we were unable to conduct statistical tests using only the individuals of a single species. We addressed both challenges by forming groups that share a common base in terms of taxonomy, and diet. Grouping multiple species together allowed for a sufficient fish count for statistical tests, enabling us to determine each groups' preferences concerning shelter sizes and distributions.

  1. size scales in all pictures please.

Size scale added to all the pictures.

  1. It is important to make clear to the readers that this could be apply when the issue is artificial reefs to coral reef fish, to restoration of coral reefs. But when the goal is fishries, this small shelters to dominant species might not function.

Thank you, we agree and we added a clarification at the beginning of the discussion that states that our research relates only to coral reef restoration and conservation, lines 515-516:

“. . Further, this design may not be suitable for fisheries or for accommodating large, non-site-attached fish.”

Reviewer 2 Report

Comments and Suggestions for Authors

Authors

It's interesting to read your manuscript, the text has good fluidity, but it still needs some fine-tuning as it still has several flaws. Please find below y suggestions.

Introduction

Line 46: “… populations, and address socio-cultural and economic values.” Throughout the entire text, this is the only time the human dimension is mentioned. This point requires the introduction of some literature justifying the need for this type of studies. Please see my suggestion for Lines 463-5.

L 77: “… and cost a lot of time,” this kind of language doesn’t seem very scientific. Please amend.

L 80: “W” is a typo. It should be “We”.

Ls: 80-88: the word “We” appears too many times. It should be more formal like: “It was designed...” “… which was placed on...” and so on.

Methods

Ls 92-4: Needs literature to support this statement/requirement.

Ls 113 onwards: Please add some literature to explain why there were used these choices. Or was it ad-hoc?

Ls 146 onwards: Lack of literature here to support the method... Please add some refs here.

Ls 160-1: Considering observational time underwater: Why not 10 minutes? What does the literature recommend?

Ls 179-81: Linked to the above. Perhaps the number of fishes was small due to very short time observations. Longer observations could improve the number of individuals using the shelter.

L 186: instead of “Excel software” it should be “Microsoft Office Excel spreadsheet” and instead of “RStudio software” it should be “R version (insert version and packages used)”. Please note that Rstudio is an integrated development environment to run R. Please add R reference at the end of MS:

R Core Team. (2021). A: A language and environment for statistical computing. R Foundation for Statistical Computing. https://www.R-project.org/

L 195: “…on their reported diet in the literature;” Which literature? Please select a few and add them.

L 200: “The species were divided into families as follows:” Either remove this sentence (as it seems redundant) or change punctuation from ":" to "."

Results

L 293: “Piscivore During” Punctuation. Needs the dash as the others before.

Discussion

L 358: “… that a year-long survey like ours” Needs better explanation : 10 months is not one year, right?

L 371: “tion[11] . Unfortunately,” 2 space typos.

L 388: “the two classes, but” 1 space typo

L 411: “night[46] . The” 2 space typos.

L 412: “still a lot we don't know,” This is not scientific language. Please amend accordingly.

L 420: “the species observed made conscious choices”. Is this correct "conscious choices"?

L 422: “process - The species” Typo: not capital letter.

L 424: “([42]; Fig. 7).” Fig 7?? The manuscript only has 5 figures???

L 446: “over 11 months.” Is the number of months correct? Please check.

L 463-5: “We conclude that when designing artificial shelters, we need to consider natural constraints such as space availability, changing structural topography, and budget constraints.”

There are 2 things here to say about this:

1- This part will be conclusions and not discussion. Therefore, it is important to add the last section: “Conclusions”.

2- The authors should also consider the human dimension that is only referred in Line 46 with nothing to justify why the RAS are useful. These are to justify that “a reef that is not useful to people is not a successful reef.” Please include important references on this aspect: Milon, Holland, and Whitmarsh (2000); Oliveira, Ramos, and Santos (2015).

Milon, J. W., Holland, S. M., & Whitmarsh, D. J. (2000). Social and economic evaluation methods. Artificial reef evaluation: with application to natural marine habitats, 165-194.

Oliveira, M. T., Ramos, J., & Santos, M. N. (2015). An approach to the economic value of diving sites: artificial versus natural reefs off Sal Island, Cape Verde. Journal of Applied Ichthyology, 31, 86-95.

Comments on the Quality of English Language

Despite some typos detected, the quality of English is good. 

Author Response

It's interesting to read your manuscript, the text has good fluidity, but it still needs some fine-tuning as it still has several flaws. Please find below my suggestions.

Introduction

Line 46: “… populations, and address socio-cultural and economic values.” Throughout the entire text, this is the only time the human dimension is mentioned. This point requires the introduction of some literature justifying the need for this type of studies. Please see my suggestion for Lines 463-5.

Thank you, we apologize it wasn’t clear but this line referred to a review paper studied by Higgins et al. (2022). “A recent study [16] evaluated the success of artificial shelters and showed that artificial reefs (AR) can provide”…

We would like to stress that artificial reefs were not the prim objective of this study. Nevertheless, we appreciate your suggestions and we added proper citations

L 77: “… and cost a lot of time,” this kind of language doesn’t seem very scientific. Please amend.

Reworded, thank you.

L 80: “W” is a typo. It should be “We”

Corrected

Ls: 80-88: the word “We” appears too many times. It should be more formal like: “It was designed...” “… which was placed on...” and so on.

Reworded

Methods

Ls 92-4: Needs literature to support this statement/requirement.

We want to thank the reviewer for this comment since it is an indication that we failed to explain the idea behind the methodology we offer here. We suggest that different designs of shelters can be used to collect the data required for the analysis we present in the paper. We do not expect others to use the specific design we used, but rather we encourage other researcher to use their own design for shelters as long as the fish are given a choice between different characteristics. We added explanations at the of the introduction- lines 93-101:

To uncover reef fish preferences for specific shelters, we focused on examining two commonly studied characteristics of reef fish shelters: size [5,27–30] and spatial distribution [34,35] Using the collected data, we hypothesized that we could determine species-specific preferences for the dominant species in the study area. We addressed the statistical challenges associated with low-volume data by employing a unique four-step analysis in our design. This analysis involved the crossed preferences of functional (diet) and systematic (family) groups, and their combination, allowing us to determine species-specific preferences. This methodology may offer a significant advantage and it may be used to unveil species-specific preferences globally, irrespective of the shelter design.

We also added an caution wording in the discussion, lines 578-581:

We would like to caution that while certain shapes, sizes, or designs may be more advantageous for some species, the same might not be the case for others [36,37]. For that reason, we encourage the examination of a variety of shelter designs to find the best fit for the target species.”

Ls 113 onwards: Please add some literature to explain why there were used these choices. Or was it ad-hoc?

Thank you. We added an explanation with references to all parts of the methods and the introduction, as some of our methods were decided based on preliminary surveys.

As for shelters design-we added a new section to the methods section

2.1.1 Experimental shelters design

Ls 146 onwards: Lack of literature here to support the method... Please add some refs here.

We added detailed explanation for the survey technique in lines 200-207. The reviewer is correct that we tailored the survey method (see reference 43) to our own situation.

Each survey commenced with the divers hovering approximately two meters from the balcony, positioned opposite to each other, and moving in opposite directions to gain an optimal total view of the area observed. Both divers documented all visible fish from this distance to minimize disturbance to the fish present. As the survey progressed, divers approached the balconies to identify and document smaller fish (Gobiidae and Blenniidae). The total duration of each observation was four minutes, as preliminary surveys at the site indicated that longer observation times during the daytime resulted in resampling of the same individuals.

.”

And for the RAS monitoring -lines 214-226

“ RAS monitoring - The monitoring of the RAS was conducted through the entrance of each shelter. Given their small size, the RAS were solely monitored by the lead diver. Observations of the RAS were not time-based, as preliminary surveys indicated that individuals would not approach the RAS when being actively observed. Additionally, we observed that individuals close to the RAS would often enter the shelter as the diver approached. The monitoring process involved an initial observation by hovering near the balcony and then circling it twice in a specific order through all the RAS. This was done to minimize disturbance to the fish. The first loop was performed approximately one meter from the balcony during day dives and 0.6 meters during night dives. During the first loop, the diver documented from the outside all the fish that passed through or inhabited the RAS. The second loop was performed in the same order, with the lead diver swimming as close to the shelters as possible to observe them through the shelter entrance [43].

Ls 160-1: Considering observational time underwater: Why not 10 minutes? What does the literature recommend?

We added an explanation for the observation time in the method section, lines 205-207:

“The total duration of the observation was four minutes since preliminary surveys in the site reveal that during the daytime- longer observation time resulted in resampling of the same individuals. During the night, the majority of the individuals positioned themselves on the balconies or in the RAS for long periods of time and were not swimming or moving around during the surveys”

Ls 179-81: Linked to the above. Perhaps the number of fish was small due to very short time observations. Longer observations could improve the number of individuals using the shelter.

As mentioned, preliminary surveys on the site reveal that longer observation time resulted in the resampling of the same individuals. During the day, the balconies and the RAS functioned as an attraction site, meaning the same individuals were observed surrounding the balconies, and moving between shelters, instead of spreading along the poles. During the night the individuals were positioned in specific locations on the balconies and inside the RAS and were not swimming, so longer observation time was not needed in any survey.

L 186: instead of “Excel software” it should be “Microsoft Office Excel spreadsheet” and instead of “RStudio software” it should be “R version (insert version and packages used)”. Please note that Rstudio is an integrated development environment to run R. Please add R reference at the end of MS:

 Amended as suggested

L 195: “…on their reported diet in the literature;” Which literature? Please select a few and add them.

Added as suggested

L 200: “The species were divided into families as follows:” Either remove this sentence (as it seems redundant) or change punctuation from ":" to "."

 Punctuation changed

Results

L 293: “Piscivore During” Punctuation. Needs the dash as the others before.

Included

Discussion

L 358: “… that a year-long survey like ours” Needs better explanation: 10 months is not one year, right?

Corrected, thank you.

L 371: "tion[11]. Unfortunately," 2 space typos.

Corrected

L 388: “the two classes, but” 1 space typo

Corrected

L 411: "night[46]. The" 2 space typos.

 Corrected

L 412: “still a lot we don't know,” This is not scientific language. Please amend accordingly.

Reworded.

L 420: “the species observed made conscious choices”. Is this correct "conscious choices"?

Reworded. Our intention was to make clear the individuals likely made an active choice and were not randomly placed in the shelters. Now appear in line 535-536;

“it appears that individuals of different species make deliberate choices, exhibiting preferences for specific features of different shelters [24,28,50].

L 422: “process - The species” Typo: not capital letter.

Corrected

L 424: “([42]; Fig. 7).” Fig 7?? The manuscript only has 5 figures???

Deleted          

L 446: “over 11 months.” Is the number of months correct? Please check

Thank you. The duration of the experiments was 11 months as shown in figure 4. Corrected.

L 463-5: “We conclude that when designing artificial shelters, we need to consider natural constraints such as space availability, changing structural topography, and budget constraints.”

There are 2 things here to say about this:

1- This part will be conclusions and not discussion. Therefore, it is important to add the last section: “Conclusions”.

Done.

2- The authors should also consider the human dimension that is only referred to in Line 46 with nothing to justify why the RAS are useful. These are to justify that “a reef that is not useful to people is not a successful reef.” Please include important references on this aspect: Milon, Holland, and Whitmarsh (2000); Oliveira, Ramos, and Santos (2015).

Thank you. We added and inserted the references accordingly.

Comments on the Quality of English Language: Despite some typos detected, the quality of English is good

Thank you

Reviewer 3 Report

Comments and Suggestions for Authors

This paper focuses on the setting of artificial fishing reefs to monitor the usage characteristics of fish on these reefs and analyze the role of fishing reefs. It has certain reference significance and is beneficial for the practical application of ecological restoration. However, there are certain issues with the writing of the article, particularly in terms of the presentation of tables and titles, which are unclear in description. Some language words are not professional enough and do not conform to English grammar habits.

1.     Figure 1. There are many ABC letters in both the large and small images in Figure 1, for example, there are letters such as ABC in Figure C, which need to be revised.

2.     Line 80, W should be the We.

3.     Line 30 should be the distribution area and quantity of coral reefs are decreasing globally, not coral reefs.

4.     Line 243, In Fig.3, The vertical axis represents the cumulative number of species, now it is inappropriate vertical axis title.  What do solid and dashed lines represent and need to be explained.

5.     Line 274, In Table 1, Cl.” stands for clumped dispersal “Sp.” stands for spread out dispersal, these two are not clear to understand. The labeling position of Families is not accurate. One blank should be left first, and Families should be placed above the names of each fish species.

6.     Due to the small number of observed samples, further verification is needed to determine whether the results obtained have statistical significance.

7.     The overall writing style is not in accordance with the format required by the journal.

8.     The discussion section is quite messy, with more than ten small paragraphs that do not focus on a few points and need to be reorganized.

9.     Need to check others problem on the article.

Comments on the Quality of English Language

need to further improve 

Author Response

This paper focuses on the setting of artificial fishing reefs to monitor the usage characteristics of fish on these reefs and analyze the role of fishing reefs. It has certain reference significance and is beneficial for the practical application of ecological restoration. However, there are certain issues with the writing of the article, particularly in terms of the presentation of tables and titles, which are unclear in description. Some language words are not professional enough and do not conform to English grammar habits.

  1. Figure 1. There are many ABC letters in both the large and small images in Figure 1, for example, there are letters such as ABC in Figure C, which need to be revised.

 Amended as suggested

  1. Line 80, W should be the We.

Corrected and reworded

  1. Line 30 should be the distribution area and quantity of coral reefs are decreasing globally, not coral reefs.

 Amended as suggested

  1. Line 243, In Fig.3, The vertical axis represents the cumulative number of species, now it is inappropriate vertical axis title.  What do solid and dashed lines represent and need to be explained.

– Amended as suggested

  1. 5.     Line 274, In Table 1, “Cl.” stands for clumped dispersal “Sp.” stands for spread out dispersal, these two are not clear to understand. The labeling position of Families is not accurate. One blank should be left first, and Families should be placed above the names of each fish species.

– Amended as suggested

  1. Due to the small number of observed samples, further verification is needed to determine whether the results obtained have statistical significance.

Thank you for your comment. We have failed to clarify that this is one of the advantages of using our methodology.  Here we show that when using low-volume data that is insufficient for statistical tests, it is still possible to determine the specific species' preferences by using our four-step analysis. We added a further explanation of the topic in

Line 491-494 in discussion

“To verify the accuracy and the alignment of the preferences we identified with the actual presence of the fish, on-site surveys were conducted. Due to the impracticality of statistical tests on individual numbers for certain species, we aggregated the abundance of each fish species across all surveys for each shelter size or dispersal type.”

  1. The overall writing style is not in accordance with the format required by the journal

Done

  1. The discussion section is quite messy, with more than ten small paragraphs that do not focus on a few points and need to be reorganized.

Thank you for this comment. We have adjusted the discussion accordingly with longer paragraphs that lead the readers through the points we wished to make.

  1.  

Comments on the Quality of English Language: need to further improve

The manuscript was edited by a professional English editor

Round 2

Reviewer 1 Report

Comments and Suggestions for Authors

You need to rewrite the hypothesis. Hypothesis is something that you will test related to a prediction.

Author Response

You need to rewrite the hypothesis. Hypothesis is something that you will test related to a prediction.

Hypotheses are now clearly written and described in the manuscript.

We would like to commend the reviewer for realizing that we had clear prediction and hypotheses on the onset of the study, but that we had to reject both of them- as the world is a complicated place.

Anyway-

In the introduction lines 98-100 now read

  • In these, we hypothesized that 1) larger shelters will host a larger number of fish species, and that 2) shelters of a similar size will host more fish species when they are clumped together as compared to when they are spread apart.

In the results section lines 374-377 now read

  • Put as a whole- we could not accept neither of our hypotheses as a general rule. Not that that larger shelters will host more fish species as compared to smaller shelters, nor that shelters of a similar size will host a more fish species when they are clumped together as compared to when they are dispersed.

In the discussion section lines 502-507 now read

  • When examining our original hypotheses, we find them somewhat naïve. Indeed, over all we could not accept neither of our hypotheses as a general rule. Not that that larger shelters will host more fish species as compared to smaller shelters, nor that shelters of a similar size will host a more fish species when they are clumped together as compared to when they are dispersed. The shelter preferences were not general across species of functional groups.

Reviewer 3 Report

Comments and Suggestions for Authors

Please further review the entire text

Comments on the Quality of English Language

Please further review the entire text

Author Response

Please further review the entire text for English quality.

The manuscript was edited again by to a professional English editor. A very extensive editing was performed as can be seen in the track- change version of the manuscript.